# Roles of Sirtuins in Hearing Protection

**DOI:** 10.3390/ph17080998

**Published:** 2024-07-28

**Authors:** Chail Koo, Claus-Peter Richter, Xiaodong Tan

**Affiliations:** 1Department of Otolaryngology-Head and Neck Surgery, Feinberg School of Medicine, Northwestern University, Chicago, IL 60611, USA; chail.koo@northwestern.edu (C.K.); cri529@northwestern.edu (C.-P.R.); 2Hugh Knowles Center for Clinical and Basic Science in Hearing and Its Disorders, Northwestern University, Evanston, IL 60208, USA; 3Department of Biomedical Engineering, Northwestern University, Evanston, IL 60208, USA; 4Department of Communication Sciences and Disorders, Northwestern University, Evanston, IL 60208, USA

**Keywords:** ototoxicity, outer hair cell, reactive oxygen species, cochlea, antioxidant, age-related hearing loss, noise-induced hearing loss, drug-induced hearing loss, apoptosis

## Abstract

Hearing loss is a health crisis that affects more than 60 million Americans. Currently, sodium thiosulfate is the only drug approved by the Food and Drug Administration (FDA) to counter hearing loss. Sirtuins were proposed as therapeutic targets in the search for new compounds or drugs to prevent or cure age-, noise-, or drug-induced hearing loss. Sirtuins are proteins involved in metabolic regulation with the potential to ameliorate sensorineural hearing loss. The mammalian sirtuin family includes seven members, SIRT1-7. This paper is a literature review on the sirtuins and their protective roles in sensorineural hearing loss. Literature search on the NCBI PubMed database and NUsearch included the keywords ‘sirtuin’ and ‘hearing’. Studies on sirtuins without relevance to hearing and studies on hearing without relevance to sirtuins were excluded. Only primary research articles with data on sirtuin expression and physiologic auditory tests were considered. The literature review identified 183 records on sirtuins and hearing. After removing duplicates, eighty-one records remained. After screening for eligibility criteria, there were forty-eight primary research articles with statistically significant data relevant to sirtuins and hearing. Overall, SIRT1 (*n* = 29) was the most studied sirtuin paralog. Over the last two decades, research on sirtuins and hearing has largely focused on age-, noise-, and drug-induced hearing loss. Past and current studies highlight the role of sirtuins as a mediator of redox homeostasis. However, more studies need to be conducted on the involvement of SIRT2 and SIRT4-7 in hearing protection.

## 1. Introduction

Sirtuins are a highly conserved family of proteins across species. Silent information regulator 2 (Sir2), the first sirtuin identified, was described as an anti-aging factor in yeasts [1]. Sir2 is a nicotinamide adenine dinucleotide (NAD^+^)-dependent histone deacetylase (HDAC) which can affect transcription in telomeres [1]. There are seven sirtuin paralogs in mammals, SIRT1-7 [2]. All mammalian sirtuins are NAD^+^-dependent HDAC with varying affinity for NAD^+^ [2,3]. They differ from the classical HDACs, HDAC1-10, which are Zn^2+^-dependent HDACs [4]. Under normal physiological conditions, NAD^+^ is important for sirtuin function and possibly a rate-limiting co-substrate [5]. NAD^+^ increases due to stresses such as exercise, which also acutely upregulates sirtuins [6,7]. Due to the requirement of NAD^+^, the functions of sirtuins have been studied in the background of cellular energy and metabolism [8]. 

In mammals, NAD^+^ is mostly recycled from nicotinamide (NAM) but can also be produced de-novo from tryptophan and niacin [9]. NAD^+^ can gain an electron to become NADH [9]. Only 10% of NAD^+^ is phosphorylated into NADP^+^, and NADP^+^ can be reduced to NADPH via catalysis by nicotinamide nucleotide transhydrogenase (NNT) [9]. The NADPH/NADP^+^ ratio is higher in the mitochondria compared to the cytosol and the nucleus [10]. This may be because NNT is a mitochondrial enzyme in the inner mitochondrial membrane [11]. NADPH is converted into NADP^+^ via glutathione (GSH) and this reduction allows the conversion of H_2_O_2_ into H_2_O and O_2_ in a redox reaction, decreasing reactive oxygen species (ROS) loads [12]. The NADPH/NADP^+^ ratio can acutely decrease due to oxidative stresses such as H_2_O_2_ and superoxide [13]. Overall, the NADPH/NADP^+^ ratio can indicate if the antioxidant defense is being maintained, and the deficiency in NAD^+^, the precursor to NADPH, can be an indication of oxidative stress in diseases such as cancers [14]. On the contrary, NAD^+^ boosting molecules such as niacin may help to remove excessive ROS and may ameliorate age-related diseases [15,16]. 

SIRT1 (mammalian Sir2) affects cell division, microtubule organization, metabolism, and transcription for many physiological processes [17]. Having an NAD binding domain, SIRT1 maintains the NADH/NAD^+^ ratio by consuming NAD^+^ [18]. Once SIRT1 binds NAD^+^, it can activate an anti-ROS pathway by deacetylating lysine residues of SOD2 [19]. SIRT1 can inhibit NF-kB, decreasing inflammatory cytokines and, subsequently, ROS [20]. SIRT2 is ubiquitously found in mammalian cells and it is an ADP-ribosyl transferase utilizing NAD^+^, which contributes to redox homeostasis [21]. SIRT2 is known to deacetylate FOXO3a, and deacetylated FOXO3a shows antioxidant properties, while acetylated FOXO3a can induce apoptosis via caspase-3 [22]. SIRT3 is expressed in the mitochondria during oxidative stress, affecting the oxidative phosphorylation in the electron transport chain and regulating ROS [23]. SIRT3 activates isocitrate dehydrogenase 2 (IDH2), a producer of mitochondrial NADPH, which activates the GSH-mediated antioxidant pathway [23]. SIRT3 also activates SOD2 and FOXO3A by deacetylation, decreasing ROS [24]. In the background of ROS and apoptosis, the role of SIRT4 has been elusive, but a recent study showed that the expressions in GSH and SOD2 decreased in *Sirt4*^−/−^ mice, suggesting that SIRT4 could be involved in activating antioxidants [25]. Like SIRT3, SIRT5 is also expressed in the mitochondria, where it activates IDH2 and mediates an anti-ROS pathway via NADPH and GSH [26]. However, SIRT3 and SIRT5 activate IDH2 differently (lysine deacetylation and desuccinylation, respectively) [24,26]. SIRT6 is known to stabilize and increase the expression of NRF2, which is known to respond to oxidative stress [27]. NAD^+^ inducers such as luteolin can act as SIRT6 activators [28]. The study on SIRT7 as an anti-ROS agent has been lacking, but a recent study showed that SIRT7 expression increased in response to treating human granulosa-lutein (hGL) cells with celastrol, a natural product with antioxidant properties [29].

Hearing loss in humans can be a sensorineural hearing loss (SNHL), resulting from cochlear damage from noise (NIHL, noise-induced hearing loss), drugs (DIHL, drug-induced hearing loss), cancer, age (ARHL, age-related hearing loss), and genetic defects [30]. Conductive hearing loss can arise due to damage in the outer and middle ear, which could be due to allergies and ear infections. One of the earliest works studying the effect of SIRT1 on ARHL included a 2014 study on ubiquinol-10 supplementation and progressive hearing threshold shifts in C57BL/6 mice [31,32]. Someya et al. have shown in C57BL/6 mouse models that SIRT3 protects against ARHL [33,34,35]. SIRT3 could exert its effect under calorie restriction and act via deacetylating IDH2 [33]. There needs to be further studies on the other members of sirtuins (SIRT2, 4, 5, 6, and 7) to establish their roles in hearing loss and protection. It is also important to establish causality between sirtuin dysregulation and hearing loss. 

There has been no review investigating sirtuins in the mechanism of hearing protection. This literature review will cover the basic principles linking sirtuins and hearing loss, categorize all peer-reviewed studies published until now, and summarize potential applications of sirtuins in ameliorating hearing loss. This review will conclude by summarizing which sirtuins are protective of hearing loss and where the research should go next. 

## 2. Selection of Studies

### 2.1. Literature Review Eligibility Criteria

The literature review aimed to collect data on the roles of sirtuins in SNHL. In May 2024, an initial literature review was conducted, using the National Center for Biotechnology Information (NCBI) PubMed search (https://pubmed.ncbi.nlm.nih.gov/, accessed on 1 May 2024). Publication date was set between 2001 and 2024. To study how sirtuins could modify hearing loss, the search terms were (sirtuin) AND (hearing), which indicated that studies must contain both words ‘sirtuin’ and ‘hearing’. Similar phrases such as sirtuin 1, hearing loss, and noise-induced hearing loss were included in the search terms. Next, a similar literature search was conducted using NUsearch, a library search tool provided by Northwestern University. The search conditions were any field, containing words ‘sirtuin’ AND ‘hearing’ in titles and abstracts, and published between 2001 and 2024. Last, ClinicalTrials.gov was checked for any ongoing clinical trials involving sirtuins in hearing loss.

The literature review further screened for in vitro and in vivo studies of cell cultures, explant cultures, animal models, and clinical studies. All eligible studies had to show links between sirtuin expression and hearing using molecular and physiological lab techniques. Therefore, studies that investigated sirtuins without consideration for hearing loss were excluded, and studies that investigated hearing loss without sirtuins as molecular targets were also excluded. Often, the search found papers that only mentioned sirtuins and hearing in the introduction and did not further discuss. Such papers were excluded. Writings without primary data, such as reviews, were excluded.

### 2.2. Identifying Bias and Heterogeneus Results

Risk of errors and bias were investigated in the studies found through the literature review. First, almost all C57BL/6 mice intrinsically develop hearing loss due to *Cdh23*^753A^ mutation starting from an early age of 2 months and develop severe hearing loss at 9 months [36,37]. In contrast, CBA/CaJ mice maintain hearing until 24 months [38,39]. Therefore, the literature search scrutinized studies using C57BL/6 mice or inbred mice to see whether the *Cdh23*^753A^ mutation could have inadvertently affected their results. One study did not use the correct controls in their Chapter 3.1 [40]. The study was supposed to compare the mice with the vehicle treatment and mice with the drug treatment, but this was not shown. Instead, conclusions were drawn by comparing the mice with no treatment and mice with the drug treatment, showing significant differences in hearing [40]. One study stated that the sirtuin expression was upregulated due to noise and sirtuin expression was inhibited by a SIRT2 inhibitor, AK-7 [41]. While the study did not show sirtuin expressions experimentally, the authors argued that AK-7 treatment decreased SIRT2 expression, subsequently decreasing sirtuin expression. There were conflicting results among studies [32,42]. In such cases, possible causes were determined by evaluating different effect measurements and different animal characteristics, and the heterogenous results were further explored in later chapters. 

### 2.3. Effect Measures in Selected Papers

Sirtuin expression was typically measured using quantitative PCR (qPCR) and the unit was the relative mRNA level of sirtuins against a reference mRNA level (e.g., beta-actin). To normalize expression levels in western blots, sirtuin expression was measured using internal controls such as glyceraldehyde-3-phosphate dehydrogenase (GAPDH). Immunostaining using conjugated antibodies/primary + secondary antibodies was typically used to visualize the localization of sirtuins in the cochlea. Immunostaining of sirtuins was often run simultaneously with myosin-VIIa to visualize hair cells, anti-CtBP2 + GluR2 to visualize synapses, and Hoechst/DAPI to visualize DNA, although each study used different staining reagents and techniques.

Auditory brainstem responses (ABRs) to pure tone stimuli at sound levels between 20 and 90 dB SPL were typically recorded as an auditory measurement in animal studies. Thresholds were determined from the recordings as the sound level for a visual response. Likewise, the delay and the amplitude of wave I were determined from the recordings. Distortion product otoacoustic emissions (DPOAEs), a measure of cochlear nonlinearity, were determined by the level of the cubic distortion product in dB SPL. Outer hair cell function is likely the dominant source for the nonlinearity in the cochlea, and a change in DPOAEs indicates a change in outer hair cell function [43].

Not all studies were considered equal in terms of certainty. More certainty was given to studies that included statistically significant expression of sirtuins, auditory testing results, and immunostaining. It was also more confident to suggest that the expression of sirtuins was causal to hearing protection if the study included sirtuin knock-out models to establish the causality of sirtuins in hearing protection.

## 3. Literature Review

NCBI PubMed search identified 81 records and NUsearch identified 102 records, resulting in 183 records in total (Figure 1). Duplicates within NCBI PubMed (*n* = 0) and duplicates within NUsearch (*n* = 24) were removed. Next, duplicates between NCBI PubMed and NUsearch were removed (*n* = 78), resulting in 81 records. Next, ineligible studies were screened. Reviews (*n* = 13) were excluded. Studies without primary data on hearing loss and sirtuin expression (*n* = 20) were excluded. Forty-eight records remained eligible for downstream discussion. SIRT1 (*n* = 29) and SIRT3 (*n* = 19) have been the most studied sirtuins, while SIRT2 (*n* = 1) and SIRT7 (*n* = 1) were rarely studied in connection to hearing loss (Figure 2). SIRT3 was the first sirtuin linked to hearing loss. Some publications had significant results in both SIRT1 and SIRT3 concerning hearing loss. The shortlisted studies from the literature review (*n* = 48) have been categorized by study type (cells, explant, and animals), treatment (small molecule drug, plant extract, and sirtuin inhibitor etc.), relevance (ABR threshold shifts and sirtuin expression etc.), and year of publication (2010–2024) (Table 1). From the shortlisted literature review papers, experimental drug candidates used in each study were categorized by the types of hearing loss they investigated (Table 2). Fourteen drugs were involved in ARHL or cellular senescence. Seven drugs were used to ameliorate noise-induced hearing loss (NIHL). Eleven drugs were investigated for their use in drug-induced hearing loss (DIHL). One drug was studied for early-onset hearing loss (EOHL). Two drugs were studied for hearing loss occurring in Ménière’s disease and one drug was studied for hearing loss occurring in diabetes type 2. All studies found investigated SNHL, including ARHL/induced cellular senescence (*n* = 25), NIHL (*n* = 9), DIHL (*n* = 11), hearing loss in Ménière’s disease (*n* = 2), and diabetes-associated hearing loss (*n* = 1). No study reported that sirtuins were involved in conductive hearing loss, auditory neuropathy spectrum disorder, or congenital deafness. In addition, no studies investigated sirtuins in connection to GJB2 mutations and other non-syndromic sensorineural hearing loss. It indicates that sirtuins most likely ameliorate cochlear damage in the inner ear, and sirtuin dysregulation can be triggered by aging and environmental factors.

## 4. Discussion

### 4.1. SIRT1 in ARHL

Of 48 studies, SIRT1 has been investigated in 29 studies. This popularity could be due to its verified role in extending life and delaying aging [86,87]. SIRT1 is a ubiquitous protein in the mammalian cochleae; it has been identified in inner hair cells (IHCs), OHCs, supporting cells (SCs), SV, and spiral ganglion neurons (SGNs) [32]. It has been suggested that SIRT1 expression in a mouse cochlea could decrease due to noise exposure, cell senescence, and ototoxic drugs, which will be discussed in later chapters (Figure 3) [51,53,56].

Xiong et al. studied the role of SIRT1 in ARHL in 2014 using C57BL/6 mice and ABR measurements [32]. It was reported that old mice (12–16 months old, *n* = 37) showed elevated ABR thresholds compared to young mice (1–2 months old, *n* = 44) at 4, 8, 16, and 32 kHz (*p* < 0.01). The old mice showed ~90% loss of OHC and ~50% loss of IHC. qPCR showed that *Sirt1* mRNA expression was ~2 times higher in old mice (*n* = 20) than young mice (*n* = 12), and this was replicated in antibody staining which showed a reduction of SIRT1 expression in old mice. The old and young mice did not differ in the endocochlear potential (EP) measurement. However, whether SIRT1 dysregulation was a causal factor in hearing loss was unclear.

In *Sirt1* KO mice (*Sirt1*^+/−^), SIRT1 deficiency delayed ARHL at 8, 16, and 32 kHz (*p* < 0.05) [42]. This was in contrast to the notion that SIRT1 can activate an anti-ROS pathway by activating SOD2 [18]. In the same study, 12-month-old *Sirt1*^+/+^ mice showed increases in hearing thresholds compared to 3-month-old *Sirt1*^+/+^ mice, while 12-month-old *Sirt1*^+/−^ mice did not show increases in hearing thresholds, showing that SIRT1 insufficiency protected against ARHL. There was no sign of loss of the SGNs and SCs in old *Sirt1*^+/−^ mice, compared to old *Sirt1*^+/+^ mice. This study tested that all mice used in the experiment had cadherin 23 mutation (*Cdh23*^753A/753A^), which is expected in C57BL/6J and causes ARHL. There was an expression of forkhead box O3 (FOXO3a) in SCs, such as Claudius and Deiters’ cells. FOXO3A is a transcriptional factor that could give rise to superoxide dismutase 2 (SOD2), which can reduce ROS, while SIRT1 can also activate SOD2 directly [51] (Figure 3). Overall, this result contrasted with the original theory that SIRT1 is a potent antioxidant and a part of the anti-apoptotic pathway, as seen in Xiong et al. [32]. 

However, all later studies indicated that SIRT1 expression is linked to hearing protection in ARHL [31,49,61,71,72,73,78]. Compared to mice younger than 2 months, mice older than 12 months showed decreased levels of SIRT1 and peroxisome proliferator-activated receptor gamma coactivator 1-alpha (PGC-1α) (*p* < 0.05) [78]. In HEI-OC1 cells, SIRT1 overexpression reduced apoptosis induced by H_2_O_2_, and increased PGC-1α expression (*p* < 0.05) [78]. This showed that SIRT1 is involved in an anti-apoptotic pathway and PGC-1α could be involved in the process of ARHL, although it was not shown whether SIRT1 overexpression would exert the same effects in vivo. In vivo protection against ARHL by SIRT1 overexpression was demonstrated using resveratrol in a separate study [73]. 

SAMP1/sku mice, originating from the senescence-accelerated mouse prone 1 (SAMP1) mouse strain, is a mouse strain displaying accelerated senility [88,89]. SIRT1 (*p* < 0.01) and SIRT3 (*p* < 0.01) expression increased in 7-month-old SAMP1/sku mice that had been given a Coenzyme Q10 (CoQ10) supplement since they were 1 month old [31]. In addition, CoQ10 also reduced hearing thresholds at 32 kHz when the mice were 7 (*p* < 0.05) and 13 months (*p* < 0.05) old. This study suggested that CoQ10 may help PGC-1α deacetylation, which is mediated by SIRT1 activation, and CoQ10 supplement may help counter ARHL in aging mice.

In 10-month-old C57BL/6 mice, the resveratrol supplement reduced ABR threshold elevations at 8, 16, and 32 kHz by 10–20 dB (*p* < 0.001) [73]. The same study showed that increased SIRT1 expression was associated with increased PTEN-induced kinase 1 (PINK1) and Parkin expression in HEI-OC1 cells under the stress of ursodeoxycholic acid, showing that SIRT1 may trigger mitophagy under oxidative stress. A similar study demonstrated that compared to 2-month-old mice, 12-month-old mice showed lower expressions of autophagy markers (LC3-II and p62) (*p* < 0.001), while small interfering RNA (siRNA)-induced reduction of SIRT1 resulted in cell deaths in HEI-OC1 (*p* < 0.05) [72]. These studies suggest that SIRT1 boosts cell survival via mitophagy and autophagy. Another study showed that β-Lapachone, an anti-cancer drug, could restore SIRT1 (*p* < 0.05) and SIRT3 (*p* < 0.05) expression to the control level in 24-month-old mice, while β-Lapachone also decreased p53 expression (*p* < 0.05) [71]. Decreased p53 expression could indicate a reduction in p53-dependent apoptosis via pro-apoptotic transcription factors such as PUMA, NOXA, BID, and BAD [90]. In a more recent study, luteolin was used as a protective agent in HEI-OC1 cells treated with hydrogen peroxide (H_2_O_2_) [61]. It reported that luteolin-induced SIRT1 expression reduced p53 (*p* < 0.05) as well as p21 expression (*p* < 0.05). While a reduced p53 expression can suggest an anti-apoptotic effect of luteolin-induced SIRT1, p21 may negatively regulate p53-dependent apoptosis [91,92]. In HEI-OC1 cells, N6-adenosine-methyltransferase 70 kDa subunit (METTL3) KO increased SIRT1 expression (*p* < 0.05) and decreased apoptotic signals (*p* < 0.05) [49]. The apoptotic signal was captured using flow cytometry during which Annexin V was used to detect apoptotic cells. However, Annexin V may bind to any cells that lost plasma membrane integrity (e.g., necrotic cells), so it may not fully represent p53-dependent apoptosis [93]. Increased SIRT1 expression did not affect METTL3 expression, indicating that SIRT1 is in a downstream signaling pathway. 

More recently, there has been an increased focus on the effect of microRNA (miRNA) and SIRT1. Currently, 16 miRNAs are known to affect SIRT1 expression [94]. It was reported that H19 overexpression could upregulate SIRT1 expression by suppressing miR-653-5p (*p* < 0.05) [58]. H19 is a long non-coding RNA (lncRNA) and its expression is typically reduced in aged mice [58]. Old mice displayed an increase in miR-29B expression (*p* < 0.05) and a decrease in SIRT1 expression (*p* < 0.05), while miR-34a inhibition restored SIRT1 expression to the control level (*p* < 0.05) [78]. Therefore, it is likely that the miRNA activity is an upstream process in the inner ear, controlling SIRT1 expression.

There were also reports of natural products such as cocoa polyphenol extract, which could increase SIRT1 and SIRT3 expression in HEI-OC1, OC-k3, and conditionally immortalized stria vascularis (SV-k1) cells under H_2_O_2_-induced oxidative stress [50]. 

### 4.2. SIRT1 in NIHL

The first electrophysiological evidence showing SIRT1 involvement in NIHL was in 2017 [76]. A resveratrol dietary supplement was used in mice to ameliorate SIRT1 reduction, which was caused by noise exposure [76]. Ginsenoside RD, a natural product, reduced noise-induced temporary threshold shift at 2, 4, and 8 kHz in mice [52,68]. The expression of BAX was elevated, and Bcl2 decreased after the noise exposure (*p* < 0.001). Pre-treatment with ginsenoside RD restored the SIRT1 expression to the original level (*p* < 0.001), while expression of BAX was reduced and B-cell lymphoma 2 (Bcl2) expression increased. Given that BAX is known to be pro-apoptotic and BCL2 is known to inhibit the release of cytochrome-c [95], it can be suggested that ginsenoside RD attenuates NIHL by inhibiting the intrinsic pathway of apoptosis. However, the study lacked the understanding of causality between SIRT1 expression and hair cell apoptosis; the study needed a method to see if ginsenoside RD would exert the same otoprotection when *Sirt1* was knocked out. Apelin-13 was another otoprotective molecule that could ameliorate SIRT1 expression in mice exposed to noise (*p* < 0.001) [65]. In a TUNEL assay, fewer apoptotic cells were seen in Apelin-13-treated mice than untreated mice after noise exposure. However, this otoprotection was not verified by ABRs or DPOAEs.

In guinea pigs, noise exposure caused NIHL, which was characterized by decreases in IHC ribbon synapse density between IHCs and SGNs, without the deaths of hair cells [57]. The number of presynaptic ribbons per IHC decreased 1 day, 1 week, and 1 month post-noise exposure (*p* < 0.01), but resveratrol-treated mice showed protection against ribbon synapse loss (*p* < 0.01). SIRT1 expression decreased by ~10 folds in SGNs (*p* < 0.01) and ~2 folds in OC (*p* < 0.01) 1 day after noise exposure, while SIRT1 expression in SV did not change. In addition, SOD2 activity decreased in response to noise exposure (*p* < 0.01) and increased in response to resveratrol (*p* < 0.01). Overall, the study gave evidence of SIRT1-mediated protection against oxidative stress in NIHL at presynaptic ribbons. 

One study showed that the expressions of SIRT1 and caspase-3 were upregulated in noise-exposed rats by approximately 0.5 folds (*p* < 0.05) [53]. This study was unique, as it indicated noise-induced apoptosis via SIRT1 expression, as opposed to other studies that stated that SIRT1 expression decreased after noise exposure and SIRT1 activates antioxidant pathways [57]. 

### 4.3. SIRT1 in DIHL

Platinum drugs and aminoglycosides have been the main drugs to be investigated in DIHL research. One of the first studies linking cisplatin and hearing loss was conducted in 2014 in which cisplatin decreased SIRT1 expression (*p* < 0.05) in HEI-OC1 cells [85]. In the same study, in NADH dehydrogenase quinone 1 (NQO1) KO mice, in which NAD^+^ metabolism is impaired, cisplatin treatment caused hearing loss at 4, 8, 16, and 32 kHz [85]. β-lapachone, known to elevate NQO1 expression, restored the hearing thresholds to pre-cisplatin treatment levels in WT mice but not in *Nqo1*^−/−^ mice. This suggested that consumption of NAD^+^ by SIRT1 may be crucial in protection against cisplatin-induced hearing loss. In a similar setting using cisplatin-treated mice, Dunnione (NAD^+^ inducer via NQO1) restored SIRT1 expression to the original level (*p* < 0.05), but not in *Nqo1*^−/−^ mice [79]. 

There has been an ongoing debate about whether SRT1720, a synthetic small molecule capable of inducing SIRT1, can be used against cisplatin-induced ototoxicity. SRT1720 was used in HEI-OC1 cells, zebrafish lateral line hair cells, and C57BL/6 mice treated with cisplatin [74]. In HEI-OC1 cells, cisplatin increased SIRT1 expression (*p* < 0.01), which contrasted with what was found previously [74,85]. At the same time, cisplatin also increased LC3-11 (*p* < 0.05) but decreased p62 expression (*p* < 0.05), while SRT1720 exacerbated these effects. This suggested that SIRT1720 may activate autophagy through SIRT1 expression. In mice, SRT1720 reduced cisplatin-induced hearing loss at 4, 8, 16, and 32 kHz (*p* < 0.01) and ameliorated hair cell deaths [74]. In a study unrelated to hearing loss, SRT1720 increased the mean lifespan of mice by 8.8% [96]. One study found that SRT1720 did not activate SIRT1 directly, and that there may be many intermediaries involved [97]. SRT2104, a more recent version of SRT1720, has gone through clinical trials for type 2 diabetes (NCT00933062, NCT00937872, NCT00938275, and NCT01031108), but its effect on hearing loss has not yet been investigated [98]. 

Thymoquinone is a natural product and an antioxidant, and it was found that thymoquinone supplement in 9-month-old C57BL/6J mice, compared to 2-month-old mice, increased SIRT1 (*p* < 0.001) and decreased Bak1 expression (*p* < 0.001) [40]. The increase in SIRT1 and decrease in Bak1 expressions show that thymoquinone increased antioxidant activity and downregulated the intrinsic pathway of apoptosis. Electron micrographs showed that thymoquinone protected the structure of stereocilia [40]. It was stated that ABR measurements showed significant threshold shifts at 8, 16, and 22 kHz in thymoquinone-treated mice compared to untreated mice, but the differences were small (~3 dB), and the experiments were not well controlled. In addition, the study did not provide evidence on whether thymoquinone-induced SIRT1 expression and reduced apoptosis were dependent or independent events.

### 4.4. SIRT3 in ARHL

In 2010, Someya et al. reported that SIRT3 could give protection against ARHL using C57BL/6 mouse models [33,34]. The study backcrossed C57BL/6J *Sirt3*^+/−^ mice four times, resulting in ~94% congenic mice of the genotypes *Sirt3*^+/+^, *Sirt3*^+/−^, and *Sirt3*^−/−^. Furthermore, 2-month-old and 12-month-old mice were referred to as young and old mice, respectively. Caloric restriction, a 25% reduction from the original, ameliorated ARHL at 8, 16, and 32 kHz (*p* < 0.05) in WTs. However, there was no significant change in hearing thresholds in *Sirt3*^−/−^ mice. This showed that SIRT3 is vital for hearing protection under caloric restriction. In addition, *Sirt3* increased IDH2 activity in the inner ear (*p* < 0.05) of *Sirt3*^+/+^ mice, but not in *Sirt3*^−/−^ mice. This showed that SIRT3 could alleviate oxidative stress by increasing IDH2 activity and upregulating NADH under caloric restriction. There was a question over the role *Cdh23*^753A^ played in ARHL of the C57BL/6J mice and how other strains would respond to caloric restriction.

Another study in cochlear cultures showed that H_2_O_2_ induced a loss of synapses and hair cells (*p* < 0.01) and reduced the expression of FOXO3a and SOD2 (*p* < 0.05) [59]. In addition, inhibition of SIRT3 exacerbated hair cell deaths, as seen in a TUNEL staining. Interestingly, 0.5 mM H_2_O_2_ increased SIRT3 expression, but 1 mM H_2_O_2_ decreased SIRT3 expression. This suggested that SIRT3 activity can increase in response to oxidative stress, but its capacity is limited and can become overwhelmed. A similar result was observed in a more recent study, which showed that SIRT3 overexpression in H_2_O_2_-treated cells could increase IDH2 expression and activate FOXO3a and SOD2. Moreover, using a TUNEL study, it was shown that SIRT3 overexpression could reduce apoptosis [54]. Western blots confirmed that the apoptosis was caused via caspase 3 [54]. 

Cytidine monophospho-N-acetylneuraminic acid hydroxylase (*Cmah*) KO mice showed hearing loss in old ages, and a microarray analysis using Expression BeadChip showed decreases of expressions in SIRT3, SIRT4, SIRT5, hypoxia-inducible factor 1 subunit alpha (Hif1α), FOXO3a, and SOD2 expression [80]. Given that HIF1α, FOXO3a, and SOD2 are downstream regulators of SIRT3, it is possible that knocking out Cmah resulted in a dysregulated SIRT3 expression and antioxidant activity. Kyoto Encyclopedia of Genes and Genomes (KEGG) pathway analysis showed enrichment in oxidative phosphorylation from the microarray data. In addition, decreased expressions were observed in SIRT3, SIRT4, and SIRT5, which are known to be expressed in the mitochondria under stress conditions [99]. Taken together, *Cmah* KO mice seemed to show mitochondrial dysregulation. This study was a rare case in which a microarray chip was used to study sirtuins and hearing loss. However, it investigated whole cochlear tissues, not single cells, and lacked auditory tests such as ABR and DPOAE in animal models.

Optic atrophy type 1 (Opa1) is a gene known for optic atrophy, in which 11 out of 19 known mutations are also associated with SNHL at various ages, from childhood to 30 years old [100]. One study found that senescent hair cells (induced by D-gal) showed increased OPA1 acetylation and decreased SIRT3 expression [46]. Kaempferol, an activator of SIRT3, increased SIRT3 expression, decreased SOD2 acetylation, decreased OPA1 acetylation, and overall increased the viability of D-gal-treated cells. The study also showed that the *Sirt3* KO genotype caused hearing loss at 5, 6, 8, 11.3, 16, 22, and 32 kHz (*p* < 0.05) in 6-month-old mice, although there was no evidence that OPA1 acetylation/deacetylation was involved in the hearing loss. Therefore, in vivo evidence that *Opa1* mutations could cause hearing loss via SIRT3 is still missing. A more recent study used *Opa1*^delTTAG^ mice, and it showed that *Opa1*^+/−^ caused hearing loss at 12 months of age (but not 1 and 6 months) in mice at 2~32 kHz (*p* < 0.01). Interestingly, 1-month-old *Opa1*^+/−^ mice had the highest beclin1 and SIRT3 expressions (*p* < 0.001) compared to 6- and 12-month-old mice. This showed that *Opa1*^delTTAG+/−^ mutant mice initially showed upregulated autophagy. Conversely, Parkin was most upregulated in 12-month-old *Opa1*^+/−^ mice compared to 12-month-old *Opa1*^+/+^ mice. This showed that mitophagy could be upregulated in older *Opa1*^+/−^ mice. There was no conclusive evidence that SIRT3 was involved in autophagy and mitophagy, but SIRT3 may have been involved in FOXO3a and PINK1/Parkin-mediated mitophagy, which could have prevented caspase-3-driven apoptosis.

There have been discussions over the relationship between Alzheimer’s disease (AD) and hearing loss [101,102]. However, no research has shown that hearing loss occurring in AD could involve dysregulation of sirtuins until recently. The model 3x*TgAD* is a widely used AD mouse model, while 3x*TgAD/Polβ*^+/−^ shows a decreased Polβ expression which causes deficient base excision repair and neuronal deaths, more closely mirroring AD occurring in humans [103]. Compared to wild type mice, 3x*TgAD/Polβ*^+/−^ mice showed hearing loss at 16–32 kHz in ABR (*p* < 0.01) and 16 kHz in DPOAE (*p* < 0.001) with a decreased NAD^+^ level in the cochlea (*p* < 0.01). In addition, 3x*TgAD/Polβ*^+/−^ mice showed a reduction in phosphoglycerate mutase 2 (PGAM2), SIRT3, and ac-SOD2 in the cochlea [44]. Given the reduction of NAD^+^, SIRT3, and SOD2 activation, there was an indication that redox homeostasis was impaired in the murine AD model.

### 4.5. SIRT3 in NIHL

NIHL was first investigated in conjunction with hearing loss in *Wld*^S^
*Tg*^+/−^
*Sirt3*^−/−^ mice [83]. *Wld*^S^
*Tg*^+/−^
*Sirt3*^−/−^ mice showed an overexpression of NAD^+^ while SIRT3 expression was abolished [83]. It was demonstrated that *Wld*^S^
*Tg*^+/−^ mice were protected from hearing loss by noise at 8, 16, and 32 kHz, but *Sirt3* KO prevented the protection [83]. NR (NAD^+^ precursor) supplement further protected against NIHL at 8, 16, and 32 kHz, but the *Sirt3*^−/−^ genotype abolished the protection [83]. NR also prevented neurite degeneration, which suggested that NR may drive antioxidant response, as neurites are typically vulnerable to ROS [104]. This suggested that SIRT3 is an upstream regulator of NAD^+^-dependent antioxidant activity in NIHL. However, the role of SIRT3 in NIHL is disputed, as one study showed that knocking out SIRT3 did not affect recovery from temporary threshold shift or synaptic loss [69]. The study suggested that *Sirt3*^−/−^ mice may have EOHL before noise exposure, but there was no clear evidence. Another study showed that the *Sirt3*^−/−^ genotype caused EOHL in 6-, 8-, and 12-week-old mice [66]. Confocal imaging showed that the *Sirt3*^−/−^ genotype caused irregular morphology of synapses in the IHCs, indicating that SIRT3 deficiency may have caused synaptopathy, an early sign of neuronal death.

Drug delivery with small biomaterials near or through the round window membrane is an important research topic in hearing loss. It has several advantages over other drug delivery methods such as injections via IV, middle ear tympanic membrane, round window membrane, and semicircular canal [105,106,107,108]. Systemic delivery using IV injections does not always reach the targets due to the blood labyrinth barrier (BLB), while intratympanic injections often cannot penetrate the round window membrane. An experimental drug, SODZIF-8, was created by embedding SOD2 and ZIF-8 (a drug delivery structure with biocompatibility and low toxicity) [55,109]. SODZIF-8 was injected in rats through the round window membrane one day before noise exposure [55]. SODZIF-8 injection reduced cochlear ROS and apoptosis, while SIRT3 and SOD2 expression increased. ABR measurements showed that SODZIF-8 reduced NIHL from day 1 after noise exposure until day 28, at frequencies between 4 and 32 kHz (*p* < 0.05). The effect of SODZIF-8 may not be limited to NIHL, as its benefit was also shown in cisplatin-induced acute kidney injury [110]. 

### 4.6. SIRT3 in DIHL

Adjudin temporarily weakens the cell adhesion between Sertoli and germ cells, and it is being researched for male contraceptives [111]. Adjudin was associated with protection against DIHL via pathways involving SIRT3 [81]. Rat cochlear explant cultures were treated with gentamicin, and the adjudin-treated group showed an increased SIRT3 expression (*p* < 0.05) and a reduction in ROS production compared to the group without adjudin treatment [81]. ABR showed a 5 to 10 dB threshold difference in gentamicin-treated mice, which were pre-treated with adjudin, compared to no pre-treatment (*p* < 0.001). Antibody staining showed the protection of OHCs in the mouse group with adjudin pre-treatment. However, a 5 to 10 dB threshold shift may not be substantial enough to warrant a future clinical trial. In another study, gentamicin treatment in HEI-OC1 cells and cochlear explants reduced SIRT3 expression [67]. Dihydromyricetin is an antioxidant known to function via PGC-1α [67,112]. Dihydromyricetin also reduced gentamicin-induced apoptosis in HEI-OC1 cells (*p* < 0.01). Interestingly, PGC-1α was found to be an upstream regulator of SIRT3 expression, as SIRT3 inhibition did not affect PGC-1α, but PGC-1α inhibition resulted in a reduction of both PGC-1α and SIRT3. This was opposed to SIRT1, an upstream regulator known to deacetylate PGC-1α in hearing protection [31]. 

Honokiol is a natural product being investigated for DIHL [66]. Honokiol is a bark isolate, like paclitaxel [113]. In mice treated with cisplatin, 1-h pre-treatment with honokiol protected hearing which was assessed by ABR, DPOAE, and immunohistochemistry counts [66]. SIRT3 expression was observed in HNK-treated mice and the protein was concentrated in OHCs, showing that honokiol may exert its antioxidant effect through SIRT3 [66]. Interestingly, honokiol treatment reduced HeLa and HCT116 cell counts by ~50%, and accelerated deaths in cells treated with cisplatin [66]. This suggested that honokiol, like sodium thiosulfate, may be a candidate as an otoprotective agent during cancer treatment. In addition, it displayed a synergistic effect with cisplatin against cancer cells, and the synergy between honokiol and anti-cancer drugs has already been observed in the past [114,115]. Honokiol has been previously known as an inducer of p53-dependent apoptosis, an inducer of both apoptosis and autophagy in cancer cells, and more recently, an inducer of paraptosis, a mode of non-apoptotic cancer cell death via mitochondrial and cytoplasmic swelling [116,117,118]. There needs to be a future study into why honokiol kills cancer cells, but not OHCs, as reported in the study by Tan et al. [66]. 

Paraquat is a known ototoxic chemical that works through ROS, causing apoptosis of IHC and OHCs [119]. In rats, both *Sirt3*^+/−^ and *Sirt3*^−/−^ mice cochleae showed hair cell loss without significant differences [63]. It suggests that SIRT3 expression does not confer protection against every ROS-generating drug, or that SIRT3-mediated otoprotection alone is not always sufficient in DIHL.

### 4.7. SIRT2 in Hearing Loss

Currently, there is little evidence to suggest that SIRT2 responds to oxidative stresses in the cochlea [50,66]. One study showed that IP injection of AK-7 (a SIRT2 inhibitor) 1 day before noise exposure resulted in IHC protection (*p* < 0.01) and ameliorated threshold shifts at 8, 16, 24, 32 kHz 1~14 days post-noise exposure (*p* < 0.05) [41]. Western blots using HEI-OC1 cells showed that SIRT2 expression increased in response to menadione, an ROS generator. However, the study did not show direct evidence that SIRT2 expression was modified in response to noise, nor if SIRT2 inhibition was causal to hearing protection. AK-7 reportedly inhibited SIRT2 in neurodegenerative mouse models in the past, but it is unknown if SIRT2 could be inhibited in the cochlea, although AK-7 is reportedly BLB-permeable [120,121]. Overall, inhibiting SIRT2 may be beneficial for treating hearing loss, contrary to SIRT1 and SIRT3.

RNA sequencing typically analyzes RNA expressions from mixed cells. Single-cell RNA sequencing (scRNAseq) gives information on individual cells. This allows the study of genome-wide gene expression by cell types such as OHCs, IHCs, and SCs [122]. Compared to microarray chips which investigate only a set number of genes, scRNAseq can detect a higher percentage of genes [80]. One available database, the Mouse Cochlea Aging Atlas (https://ngdc.cncb.ac.cn/aging/single-cell_marker?project=Mouse_Cochlea, accessed on 1 May 2024), compiled the list of mRNAs which were sequenced by a 2023 scRNAseq project on ARHL [123]. The Atlas comparing old (15-month-old) and young (2-month-old) C57BL/6J mice showed SIRT2 logFC of 1.13 (FDR adj *p* = 5.23 × 10^−263^) in Deiters’ cells/outer pillar cells and logFC of 1.66 (FDR adj *p* = 2.54 × 10^−71^) in oligodendrocytes of the cochlear nerves. SIRT2 upregulation in aging mice was consistent with a previous study which showed that SIRT2 increased under oxidative stress in HEI-OC1 cells [41]. However, RNAseq did not capture the expression of the other sirtuin paralogs (SIRT1 and SIRT3-7). The rest of the cell types such as hair cells, SGNs, intermediate cells, and Reissner’s membrane cells also lacked data on the expressions of sirtuins. Genes involved in hearing loss, *Sod2*, *Pink1*, and *Idh2*, were searched in the sequencing results, but failed to be sequenced in any cell type. There could be several reasons; scRNAseq could be affected by low read depth (low coverage) and low-quality RNA samples (low RIN), and the lengths of transcripts could cause enrichment bias. Therefore, it would not be possible to read the entire genome equally, and some sirtuins may not have been mapped in the Mouse Cochlea Aging Atlas. One solution would be targeting relevant genes only by using custom panels of genes and increasing their read depth [124]. For example, 125 genes are currently associated with nonsyndromic hearing loss [125], and it would be promising to design a panel using the SureSelect Target Enrichment System to target the 125 genes.

### 4.8. SIRT7 in Hearing Loss

SIRT7 has been rarely studied in connection to hearing loss [84]. SIRT7 is unique among sirtuins in that it is mostly expressed in the nucleolus and activates RNA polymerase I via deacetylation of cyclin-dependent kinase 9 (CDK9) [126,127]. One study showed that SIRT7 was involved in the deacetylation of GA binding protein β1 (GABPβ1), an important transcription factor for mitochondrial genes [84]. *Sirt7*^−/−^ mice displayed ARHL at 33–34 weeks compared to 8–9 weeks at 5, 8, 10, 15, and 30 kHz. *Sirt7*^+/+^ mice also showed the same hearing loss at 30 kHz, but not at 5, 8, 10, and 15 kHz. This showed that hearing loss observed in *Sirt7*^−/−^ mice could just have been due to aging. C57BL/6J mice used in this experiment likely harbored *Cdh23*^753A^ mutation causing ARHL, but it was not genotyped. In the same study using HEK293T cells, overexpression of SIRT7 decreased the level of ac-GABPβ1, while SIRT1 and SIRT6 did not, in western blot assays. Overall, SIRT7 deficiency seen in *Sirt7*^−/−^ mice suggested that it may accelerate ARHL through mitochondrial dysregulation, but there should be further studies with mice strains which maintain normal hearing over the lifespan.

### 4.9. SIRT4, SIRT5, and SIRT6

Little evidence suggests that SIRT4, SIRT5, and SIRT6 could be involved in hearing loss. Only one study so far suggests their involvements in hearing loss, albeit without evidence of auditory tests [128]. The study in ARHL showed the distribution of SIRT1-7 in young (8 weeks old) and old (22-month-old) CBA/J mice using quantitative reverse transcription PCR (RT-qPCR) [128]. SIRT1, SIRT2, SIRT4, SIRT5, SIRT6, and SIRT7 showed an increase in expression in vestibular end organs (utricle, saccule, and crista ampullae) of old mice (*p* < 0.05). SIRT1, SIRT2, SIRT4, SIRT6, and SIRT7 showed decreased expression in vestibular ganglions. SIRT1, SIRT3, and SIRT5 showed decreased expression in cochleae (OC, lateral walls, and SGNs).

In the same study, SIRT3 was the least expressed sirtuin in all tissues, in young and old mice, suggesting that SIRT3 expression may be stress-dependent [128]. In addition, SIRT3 did not show much change between young and old mice. SIRT2 and SIRT4 expression showed a very similar, almost equal expression in vestibular ganglions, cochleae, and acoustic nerves. SIRT5 showed a slight decrease in expression in the cochlea of aging mice, and this showed that SIRT5 may have similar roles to SIRT3, which is associated with ARHL. This is supported by past studies showing that IDH2 is a common substrate of SIRT3 and SIRT5, albeit with different modes of binding [47,129]. SIRT6 was sparsely expressed in various inner ear tissues and did not differ much in young and old mice [128]. SIRT7 was the most expressed sirtuin in the cochlea, although there was no difference in young and old mice. This study would have benefitted from physiological auditory tests, SIRT KO mouse models, and inhibiting transcription factors upstream and downstream of sirtuins to ascertain the causality between sirtuin expressions and hearing. In addition, a whole-genome analysis could have been an effective tool to verify the roles of sirtuins and hearing [80].

## 5. Summary

Sirtuins are a family of NAD^+^-dependent deacetylases working through activating key enzymes for stress resistance pathways, and are the key regulators of the intrinsic anti-ROS systems. With seven highly conserved members (SIRT1-7) which differ in catalytic activities and subcellular locations (in cytoplasm: SIRT1 and SIRT2; in mitochondria: SIRT3-5; and in nucleus: SIRT1, 2, 6, 7), sirtuins govern cellular processes including homeostasis, responses to stress, DNA damage repairing, inflammation, and apoptosis. Since the indication of SIRT3 involvement in ARHL in calorie-restricted mice in 2010, there has been an increase in research in SIRT1 and SIRT3 in relation to hearing protection. Recently, there have been probes into other sirtuins such as SIRT7, studied in 2014, and SIRT2, studied in 2019. It is expected that more roles of sirtuins will be found in future studies.

SIRT1 acts in an anti-inflammatory pathway via NF-κB, resulting in an anti-apoptosis response. In ARHL, SIRT1 can act via LC3-I and LC3-II to increase autophagy and mitophagy, keeping cells less sensitive to apoptosis and allowing them to overcome death. SIRT1 can also act via activating FOXO3a and SOD2 to keep redox in balance against extra ROS which can be caused by noise exposure. SIRT1-mediated redox homeostasis is important, as ROS can activate caspase-3-mediated apoptosis and hair cell death and neuronal death in the cochlea. Aging and AD negatively regulate the expression of SIRT1 and SIRT3 in the cochlea, while specific mouse models such as OPA1^del^ also show aberrant SIRT3 expression. Unlike SIRT1, SIRT3 seems dependent on oxidative stresses for it to be expressed. One example is caloric restriction, which can activate IDH2 and ensure the redox balance by maintaining the oxidized-to-reduced-NAD (NAD^+^/NADH) ratio, keeping ROS in check over time, and delaying ARHL. SIRT3 and SIRT1 share specific regulatory proteins, such as SOD2 and FOXO3A, showing that they are both capable of diffusing superoxide and preventing caspase-3-mediated apoptosis. In addition, certain stresses such as noise and ototoxic drugs initially increase SIRT1 and SIRT3 concentrations, but the concentrations sharply decrease afterward. Similarly, SIRT1 and SIRT3 were linked to ARHL, NIHL, and DIHL. However, there are several differences. One difference is that only SIRT3 affects cisplatin-induced hearing loss. The role of SIRT1 in cisplatin-induced hearing loss is still to be confirmed. In addition, flavonoids and natural products increase the expression of SIRT3, but not SIRT1. This suggests that SIRT3 is the primary inhibitor of ROS caused by ototoxic drugs. Another important difference is that the *Sirt1*^−/−^ mouse genotype is more punishing than the *Sirt3*^−/−^ mouse genotype. Most of the mice missing SIRT1 (*Sirt1*^−/−^) die before birth, while mice missing SIRT3 (*Sirt3*^−/−^) survive. Studies investigating intrinsic hearing loss in SIRT3^−/−^ are limited. It implies that SIRT3 as an antioxidant mediator is required but not essential. SIRT7 is special in that it is expressed in the nucleolus, but mediates the expression of mitochondrial proteins. Like *Sirt1*^−/−^ mice, C57BL/6 *Sirt7*^−/−^ mice show ARHL, possibly via mitochondrial dysregulation. However, the fact that *Sirt7*^+/+^ mice also lose hearing casts doubt over the result, and it is suggested that another study is to be conducted in the future with mice strains which can maintain normal hearing over the lifespan. SIRT2, the odd one out, seems to ameliorate NIHL when inhibited. However, there is doubt over the effectiveness of the SIRT2 inhibitor (AK-7) and it is hoped that a more effective SIRT2 inhibitor will be found in the future. A study involving *Sirt2*^−/−^ mice would be a good follow-up study to show evidence that SIRT2 is required to control noise-induced ROS. SIRT4, SIRT5, and SIRT6 lacked any evidence of involvement in hearing loss, but their expressions were slightly altered in an ARHL model of mice in various tissue types of an ARHL model of mice. The lack of studies found in SIRT4-6 is a major limitation of this review, as it is unclear if SIRT4-6 are indeed unrelated to the mechanisms of hearing loss, or if SIRT4-6 simply lack the research volumes. A whole-genome assay such as scRNAseq and SureSelect Target Enrichment System would be beneficial in discovering the roles of SIRT4-6 and other transcriptional regulators involved in hearing loss.

## Figures and Tables

**Figure 1 pharmaceuticals-17-00998-f001:**
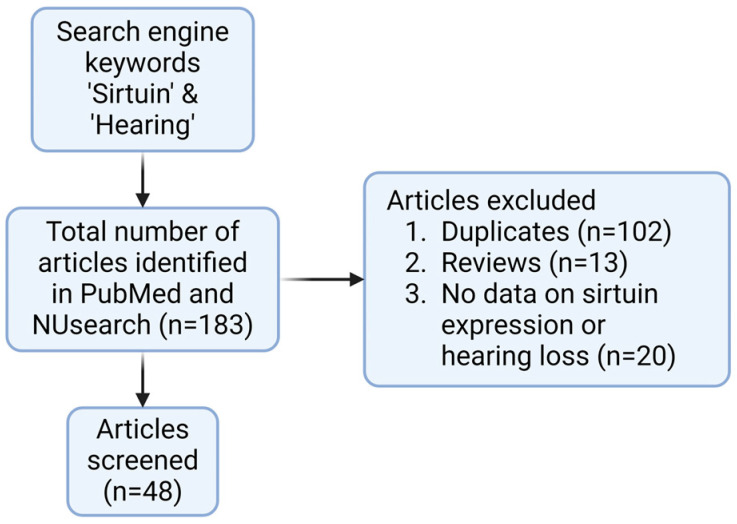
Flow diagram representing the screening process for articles investigating sirtuins in hearing loss. Created with BioRender.com.

**Figure 2 pharmaceuticals-17-00998-f002:**
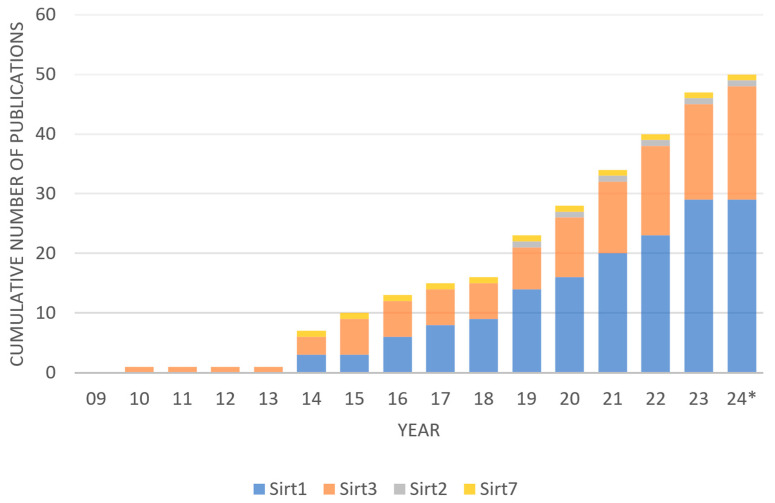
Sirtuin and hearing loss research over time by the number of publications in each sirtuin paralog * Up to May 2024.

**Figure 3 pharmaceuticals-17-00998-f003:**
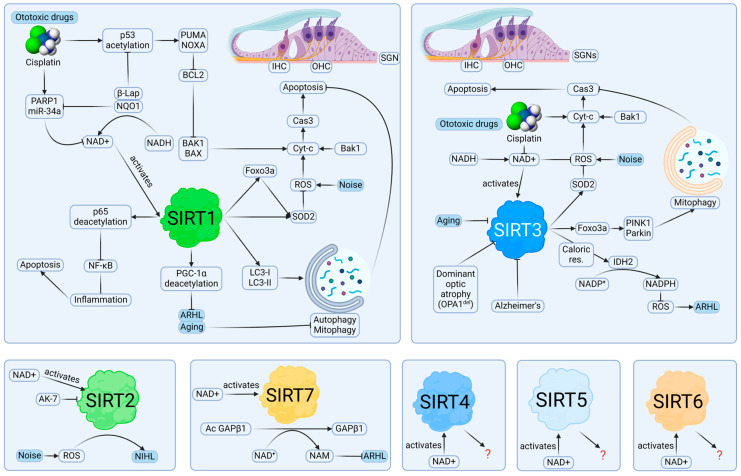
Potential roles of sirtuins in hearing loss. SIRT1 and SIRT3 can modulate ROS in the pathologies leading to hearing loss, evidenced by studies in cell cultures and animal models. The major mechanism is the inhibition of intrinsic apoptosis pathway in IHCs, OHCs, and SGNs. Autophagy and mitophagy, which are triggered by pathways involving SIRT1 and SIRT3, can keep the cells under the threshold of caspase-3 activity sufficient for triggering apoptosis, providing protective roles. Inhibition of SIRT2 may be beneficial in NIHL. It is unclear if SIRT4, SIRT5, and SIRT6 are involved in hearing loss or hearing protection. Created with BioRender.com.

**Table 1 pharmaceuticals-17-00998-t001:** All-time studies linking sirtuins and hearing loss listed chronologically.

Study Model *	Treatment	Sirtuin	Relevance to Sirtuins in Hearing Loss
Alzheimer’s mouse model (TgAD) with EOHR (2024) [44]	None	*Sirt3*	*Sirt3* mRNA decreased in male mice (*p* < 0.05)
Dominant optic atrophy mouse model (Opa1^delTTAG^) with ARHL (2024) [45]	None	*Sirt3*	SIRT3 (western blot) expression increased in heterozygote KO (*p* < 0.001)
HEI-OC1 cells with senescence induced by D-gal (2024) [46]	Kaempferol; a SIRT3 activator	*Sirt3*	Kaempferol increased SIRT3 expression (*p* < 0.001)
24-month-old *Sirt3*^−/−^ mouse model (2024) [47]	None	*Sirt3*	Mitochondrial structural damage occurred in *Sirt3*^−/−^*. Sirt3*^−/−^ genotype resulted in hearing loss at 5~32 kHz (*p* < 0.05) assessed by ABRs
Ménière’s disease biomarker study in clinics (2023) [48]	Coriolus mushroom; an antioxidant	*Sirt1*	A small increase in SIRT1 in Ménière’s disease patients, a further increase after 6 months Coriolus treatment
OC1 cells (2023) [49]	METTL3, a methyltransferase	*Sirt1*	METTL3 KO decreased SIRT1 expression and increased apoptosis (*p* < 0.05)
Senescent HEI-OC1, OC-k3, SV-k1 cells treated with H_2_O_2_ (2023) [50]	Cocoa polyphenols extract	*Sirt1* *Sirt3*	Cocoa polyphenols extract increased SIRT1 and SIRT3 expression (ELISA) in all three types of auditory cells (*p* < 0.05)
Senescence induced by D-gal in strial marginal cells (SMCs) (2023) [51]	Adenovirus-Sirt1	*Sirt1*	Adenovirus-Sirt1 returned the expression of SOD2 to the level before D-gal induction (*p* < 0.05)
Guinea pigs exposed to helicopter noise (2023) [52]	Ginsenoside RD	*Sirt1*	Noise exposure decreased SIRT1 expression in the cochlea (*p* < 0.05)
Rats exposed to noise (2023) [53]	None	*Sirt1*	*Sirt1* mRNA decreased in noise-exposed rats (*p* < 0.05)
Cochlea basilar membrane cells (2022) [54]	H_2_O_2_	*Sirt3*	NR reduced H_2_O_2_-induced apoptosis of hair cells and IHC synapses due to H_2_O_2_ possibly through SIRT3 expression
Rats exposed to noise (2022) [55]	SODZIF8; superoxide dismutase nanoparticle	*Sirt3*	SIRT3 expression decreased after noise exposure, but SODZIF8 counteracted this effect as shown in a quantitative western blot (*p* < 0.05)
Mice ototoxicity induced by aminoglycosides (2022) [56]	dihydronicotinamide riboside (NRH); a reduced form of NR	*Sirt1*	SIRT1 expression increased in kanamycin-treated mice after NRH IP injection (*p* < 0.01), and the injection lowered hearing thresholds (*p* < 0.01)
Guinea pigs exposed to noise (2022) [57]	Resveratrol	*Sirt1*	Resveratrol increased SIRT1 expression in noise-exposed guinea pigs (western blot), but unclear hearing threshold change
HEI-OC1 cells given oxidative stress by H_2_O_2_ (2022) [58]	H19; an LncRNA which is downregulated in old mouse cochleae	*Sirt1*	H_2_O_2_ increased H19 and SIRT1 expression (*p* < 0.05). H19 overexpression under H_2_O_2_ increased SIRT1 expression and reduced ROS (*p* < 0.05)
OC cells treated with H_2_O_2_ (2022) [59]	3-TYP; a SIRT3 inhibitor	*Sirt3*	H_2_O_2_ induced loss of synapses and hair cells (*p* < 0.01) and 3-TYP further induced loss of OHCs only (*p* < 0.05)
74-week-old rats using NAD^+^ level to measure sirtuins (2021) [60]	Environmental enrichment with plastic toys	*Sirt1*	Sirt1 activity was higher in the enrichment group, and the ABR click threshold was lower in the enrichment group
8-month-old C57BL/6J mice (2021) [40]	Thymoquinone; an antioxidant	*Sirt3*	Thymoquinone lowered hearing thresholds (8, 16, 22 kHz), increased SIRT1 expression (*p* < 0.001), and reduced Bak1 expression (*p* < 0.001)
HEI-OC1 cells treated with H_2_O_2_ (2021) [61]	Luteolin; an antioxidant flavonoid	*Sirt1*	Luteolin reduced H_2_O_2_-induced senescence (*p* < 0.05) via upregulating SIRT1 (*p* < 0.05)
OC explants extracted from C57BL/6 mice treated with cisplatin (2021) [62]	TES-1025; a promoter of NAD^+^	*Sirt1*	Cisplatin treatment with TES-1025 increased SIRT3 expression compared to cisplatin-only (*p* < 0.05), and antibody staining showed that TES-1025 protected hair cells
Sirt3 KO mice cochlear explants exposed to ototoxic herbicide (2021) [63]	Paraquat; a superoxide generator	*Sirt3*	Paraquat destroyed hair cells, but different *Sirt3* genotypes did not cause any difference
3-month-old male B6 mice (2021) [64]	MNAM; a possible SIRT1 inducer	*Sirt1*	High-fat diet reduced SIRT1 expression (*p* < 0.01), but MNAM returned it to the control level (*p* < 0.01)
Mice exposed to noise (2021) [65]	Apelin-13; a neuroprotective agent	*Sirt1*	Apelin-13 increased SIRT1 expression in noise-exposed cochlea (*p* < 0.001)
Mice exposed to cisplatin (2020) [66]	Honokiol	*Sirt3*	Honokiol protected hearing as shown through reduced threshold elevations in ABR (*p* < 0.01) and reduced amplitudes in DPOAE (*p* < 0.01). Antibody staining showed increased SIRT3 expression in OHCs in honokiol-treated mice (*p* < 0.01)
Gentamicin-treated HEI-OC1 cells and cochlear explants (2020) [67]	Dihydromyricetin	*Sirt3*	Gentamicin reduced SIRT3 expression which returned to normal upon dihydromyricetin induction (*p* < 0.001)
Male guinea pigs exposed to a loudspeaker (2020) [68]	Ginsenoside RD	*Sirt1*	Noise exposure decreased SIRT1 expression in the cochlea (*p* < 0.001), but GSRd returned it to the normal level (*p* < 0.01)
Sirt3 KO Mice exposed to noise (2020) [69]	None	*Sirt3*	Sirt3 loss does not affect recovery from temporary threshold shift or synaptic loss but may affect the pre-noise exposure threshold
Ménière’s disease patients with SNHL (2019) [70]	Coriolus mushroom; an antioxidant	*Sirt1*	Increase in SIRT1 in Ménière’s disease patients after Coriolus treatment (*p* < 0.05)
24 months old C57BL/6J mice with ARHL (2019) [71]	β-lapachone; an anti-cancer drug	*Sirt1* *Sirt3*	β-lapachone restored SIRT1 activity to the control level in 24 months old mice (*p* < 0.05) and partially restored SIRT3 activity (*p* < 0.05) and reduced hearing thresholds at 4, 8, 16, 32 kHz (*p* < 0.05)
12 months old C57BL/6 mice with less SIRT1 due to age (2019) [72]	Resveratrol dietary supplement, SIRT1 siRNA	*Sirt1*	12-month-old mice had a lower SIRT1 expression (*p* < 0.001), and lower autophagy markers (LC3-II and p62) than 2-month-old mice (*p* < 0.001). Resveratrol reduced ABR threshold elevations in the aging mice (*p* < 0.05)
10 months old C57BL/6 mice (2019) [73]	Resveratrol supplement	*Sirt1*	Resveratrol reduced ABR threshold elevations at 8, 16, and 32 kHz (*p* < 0.001)
HEI-OC1 cells treated with cisplatin (2019) [74]	SRT1720; SIRT1 inducer 3-MA; autophagy inhibitor	*Sirt1*	Cisplatin increased SIRT1 expression (*p* < 0.01), and SRT1720 increased LC3-11 (*p* < 0.05) but decreased p62 expression (*p* < 0.05). Autophagy inhibition prevented SRT1720-mediated cell survival (*p* < 0.005)
Zebrafish lateral line hair cells treated with cisplatin (2019) [74]			SRT1720 increased the number of lateral line cells after 24 h of cisplatin treatment compared to controls (*p* < 0.005)
C57BL/6 mice treated with cisplatin (2019) [74]			SRT1720 reduced cisplatin-induced ABR threshold elevations at 4, 8, 16, and 32 kHz (*p* < 0.01) and reduced hair cell loss
C57BL/6J mice exposed to noise (2019) [41]	AK-7 (SIRT2 inhibitor) IP injection 1 day before noise exposure	*Sirt2*	SIRT2 inhibition resulted in IHC protection (*p* < 0.01) and reduced ABR threshold elevations at 8, 16, 24, and 32 kHz (*p* < 0.05)
Sprague Dawley rats with hearing loss induced by 3-NP (2018) [75]	EGb 761; otoprotective leaf extract	*Sirt1*	EGb 761 reduced ABR threshold elevations at 8, 16, and 32 kHz. EGb 761 preserved the number of SIRT1-positive type II fibrocytes compared to controls
C57BL/6 mice exposed to noise (2017) [76]	Resveratrol dietary supplement before noise exposure	*Sirt1*	Resveratrol attenuated cochlear SIRT1 decrease upon noise exposure
Diabetic db/db mice (obese) versus db/m mice (lean) (2017) [77]	miR-34a inhibitor	*Sirt1*	SIRT1 expression was lower and HIF-1α was higher in db/db mice (*p* < 0.05). MiR-34a inhibition increased SIRT1 expression (*p* < 0.05). db/db mice showed higher threshold shifts between 0.5 and 32 kHz (*p* < 0.05)
C57BL/6. Young mice (<2 months old) and old mice (12< months old) (2016) [78]	miR-29b inhibitor	*Sirt1*	Old mice showed decreased SIRT1 and increased miR-29B expression (*p* < 0.05). miR-34a inhibition restored SIRT1 expression to the control level (*p* < 0.05)
*Sirt1*^+/−^ mice (2016) [42]	None	*Sirt1*	Sirt1 deficiency delayed AR at 8, 16, and 32 kHz (*p* < 0.05) and protects hair cells and SGNs
Male C57BL/6 mice with the *Nqo1*^−/−^ genoype treated with cisplatin (2016) [79]	Dunnione; an NAD^+^ inducer acting via NQO1	*Sirt1*	Dunnione restored SIRT1 expression previously decreased by cisplatin treatment (*p* < 0.05), but not in *Nqo1*^−/−^ mice
Microarray analysis on Cmah KO mice, which are linked to ARHL (2015) [80]	None	*Sirt3*	SIRT3, SIRT4, SIRT5, Hif1α, SOD2 expressions were reduced in Cmah KO mice
Cochlear explant cultures treated with gentamicin (2015) [81]	Adjudin	*Sirt3*	Adjudin increased SIRT3 expression (*p* < 0.05)
C57BL/6. Young mice (<2 months old) and old mice (12< months old) (2015) [82]	Resveratrol	*Sirt1*	Aging increased miR-34a and p53, decreased SIRT1 (*p* < 0.05)
*Wld*^S^*Tg*^+/−^*Sirt3*^−/−^ mice (overexpression of NAD^+^ with *Sirt3* deletion) exposed to noise (2014) [83]	NR	*Sirt3*	*Wld*^S^*Tg*^+/−^ mice are protected from NIHL (*p* < 0.001), but *Sirt3*^−/^^−^ genotype abolishes the protection (*p* < 0.001) at 8, 16, and 32 kHz. NR reduces NIHL depending on *Sirt3*^−/−^ genotype at 8, 16, 32 kHz (*p* < 0.005). NR prevents neurite degeneration
*Sirt7*^−/−^ mice (2014) [84]	None	*Sirt7*	*Sirt7*^−/−^ mice showed hearing loss at 4, 8, 16, and 32 kHz at 33 weeks compared to *Sirt7*^+/+^ mice (*p* < 0.05)—no difference at 8 weeks
HEI-OC1 cells NQO1 KO mice (2014) [85]	β-lapachone	*Sirt1*	Cisplatin decreased SIRT1 expression (*p* < 0.05), while β-lapachone restored the SIRT1 expression (*p* < 0.05) in cells. β-lapachone restored hearing thresholds at 4, 8, 16, and 32 kHz to pre-cisplatin treatment level in mice (*p* < 0.05), but this was not seen in NQO1 KO mice
C57BL/6 mice which are young (1–2 months) and old (12–16 months) (2014) [32]	None	*Sirt1*	Hearing loss occurs in old mice at 4, 8, 16, and 32 kHz (*p* < 0.01) and hair cell loss is seen in immunostaining (*p* < 0.05). SIRT1 expression was lower in cochlea (*p* < 0.05), but not expressed in auditory cortex
SAMP1/Sku Slc mice (develop ARHL) were treated from young (1 month), middle (7 months), and old (13 months) (2014) [31]	Ubiquinol-10 (CoQ10)	*Sirt1* *Sirt3*	SIRT1 (*p* < 0.01) and SIRT3 (*p* < 0.01) expression increased in 7-month-old mice which had been given CoQ10 supplement since they were 1 month old. CoQ10 also reduced hearing thresholds at 32 kHz when the mice were 7 (*p* < 0.05) and 13 months (*p* < 0.05) old
C57BL/6J *Sirt3*^+/−^ mice backcrossed 4 times (~94% congenic), 2 months or 12 months old (2010) [33]	Caloric restriction, 25% reduction from the original	*Sirt3*	In 12-month-old *Sirt3*^+/+^ mice, caloric restriction lowered thresholds at 8, 16, and 32 kHz (*p* < 0.05). In 12 months old Sirt3^−/−^ mice, there was no significant change

* Studies were listed chronologically beginning with the most recent study at the top.

**Table 2 pharmaceuticals-17-00998-t002:** Drugs investigated in relation to different types of hearing loss.

Drug Name *	Identity	Sirtuin	Mechanism	Drug Target	Subject Tested	Ref
Kaempferol	Polyphenol	*Sirt3*	OPA1 deacetylation and SIRT3 activation	ARHL	Cells	[46]
Cocoa polyphenol extract	Polyphenol	*Sirt1* *Sirt3*	Inhibits auditory cell senescence through increasing SIRT1, SIRT3	ARHL	Cells	[50]
Adenovirus-Sirt1	Drug vector	*Sirt1*	Releasing DNA, which translocates to the nucleus	ARHL	Cells	[51]
NR	Nucleoside	*Sirt3*	Reduction of hair cell apoptosis via increasing SIRT3	ARHL	Cells, mice	[54,83]
H19	LncRNA	*Sirt1*	Inhibition of ROS via inhibition of miR-653-5p and increase in SIRT1	ARHL	Cells	[58]
Thymoquinone	Natural quinone product	*Sirt3*	Increase in SIRT3 and decrease in Bak1-mediated apoptosis in cochleae	ARHL	Mice	[40]
Luteolin	Polyphenol	*Sirt1*	Reducing cellular senescence via an increase in SIRT1 and inhibition of p53	ARHL	Cells	[61]
MNAM	Metabolized NAM	*Sirt1*	Possible SIRT1 inducer	ARHL	Mice	[64]
β-lapachone	Natural quinone product	*Sirt1* *Sirt3*	Increase in NAD^+^ through SIRT1, SIRT3	ARHL	Cells, mice	[71,85]
siRNA SIRT1	siRNA drug	*Sirt1*	RNAi	ARHL	Mice	[72]
miR-34a	miRNA	*Sirt1*	Binding to mRNA to repress translation	ARHL	Mice	[73]
miR-29b inhibitor	Oligo-nucleotide	*Sirt1*	Antisense oligonucleotides complementary to their targets, binding target miRNA which then are unable to bind their original target	ARHL	Mice	[78]
Ubiquinol-10 (CoQ10)	Coenzyme produced in the body	*Sirt1* *Sirt3*	Involved in the mitochondrial electron transport chain	ARHL	Mice	[31]
Ginsenoside RD	Polyphenol	*Sirt1*	Upregulation of SIRT1 and SOD2	NIHL	G. pigs	[52,68]
SODZIF8 (superoxide dismutase nanoparticle)	Superoxide dismutase mimic	*Sirt3*	Increase in SIRT3 via SODZIF8 and reducing ROS	NIHL	Rats	[55]
Resveratrol	Polyphenol	*Sirt1* *Sirt3*	Reduces ROS in NIHR and DIHL through SIRT1 and SIRT3 pathways	NIHL	G. pigs, mice	[57,76,81]
Apelin-13	Oligopeptide	*Sirt1*	Decreasing caspase-3 and BAX via increasing SIRT1	NIHL	Mice	[65]
AK-7	Small molecule drug	*Sirt2*	SIRT2 inhibitor	NIHL	Mice	[41]
NRH	Reduced form of nicotinamide riboside	*Sirt1*	Sirt1 activation and protection against drug ototoxicity	DIHL	Mice	[56]
TES-1025	Pyrimidine derivative	*Sirt1*	ACMSD inhibitor which promotes NAD^+^ via SIRT1	DIHL	Mice	[62]
Honokiol	Polyphenol	*Sirt3*	SIRT3 activation, causing IDH2 deacetylation and ROS reduction	DIHL	Mice	[66]
Dihydromyricetin	Polyphenol	*Sirt3*	Protection against ototoxicity via PGC-1a/SIRT3 signaling	DIHL	Cells	[67]
β-lapachone	Natural quinone product	*Sirt1* *Sirt3*	Increase in NAD^+^ through SIRT1 and SIRT3 expressions	DIHL	Cells, mice	[71,85]
SRT1720	Small molecule drug	*Sirt1*	SIRT1 inducer	DIHL	Cells, mice	[74]
3-MA	Adenine drug	*Sirt1*	Autophagy inhibitor	DIHL		
EGb 761	Leaf extract	*Sirt1*	Antioxidant	DIHL	Rats	[75]
Dunnione	Leaf extract	*Sirt1*	NAD^+^ inducer via NQO1	DIHL	Mice	[79]
Adjudin	Indazole-carboxylic acid	*Sirt3*	Reversible germ cell loss	DIHL	Mice cochlear explant	[81]
METTL3	Methyl-transferase	*Sirt1*	Decreasing ROS and apoptosis through increasing SIRT1	EOHL	Cells	[49]
Coriolus mushroom	Poly-saccharide	*Sirt1*	Increasing SIRT1 and neuron protection	Ménière’s disease	Clinical	[48,70]
miR-34a inhibitor	Oligo-nucleotide	*Sirt1*	Antisense oligonucleotides complementary to their targets, binding target miRNA which then are unable to bind their original target	Diabetes Type 2	Mice	[77]

* Studies were listed chronologically beginning with the most recent finding at the top, and then filtered by drug target.

## Data Availability

Data sharing is not applicable.

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
