# Peer review of "Roles of Sirtuins in Hearing Protection"

_pharmaceuticals, 2024, doi:10.3390/ph17080998_

Round 1

Reviewer 1 Report

Comments and Suggestions for Authors

The mammalian sirtuins family, SIRT1–SIRT7, are implicated in a variety of cellular functions and have got an increasing interest from a therapeutic point of view. The wide range of cellular activities of the sirtuins suggests that they could constitute therapeutic targets to combat metabolic, proliferative and neurodegenerative/age-related diseases, among which hearing loss is one. Thus SIRT1 has been suggested to facilitate neuronal survival in e.g. the inner ear.

The present review is the first to explore the scientific background of sirtuins and possible protection of various types of sensorineural hearing loss caused by aging, noise and drugs. A search regarding “sirtuin” and “hearing” in titles and abstracts was performed. In total 183 records were obtained. After removal of duplicates, reviews and studies without primary data on hearing loss and sirtuin expression, 48 peer-reviewed studies remained for analysis.

The material is well structured and the various sirtuins – 1-7 – and their mode of action in relation to hearing loss are carefully described. The review concludes by summarizing which sirtuins are protective of hearing loss and suggests which future research should be performed. 

SIRT1 and SIRT3 seems to be linked to aging, noise and drugs. It appears that only SIRT3 affects cisplatin-induced hearing loss. The role of SIRT1 in cisplatin-induced hearing loss is still to be confirmed. The role of SIRT2 is less obvious. It seems to improve noise-induced hearing loss when inhibited. SIRT7 has been rarely studied in connection to hearing loss but there is some results indicating that  SIRT7 deficiency may accelerate age-related hearing loss. When it comes to SIRT4, SIRT5 and SIRT6 all of them so far lack any evidence of involvement in hearing loss.

To summarize this sirtuin-review is well performed and analysed. The results are interesting but still further research is needed to appoint the role of surtuins in cochlear pathophysiology. I strongly suggest this review for publication in Pharmaceuticals in its present form.

Author Response

Comment: 

I strongly suggest this review for publication in Pharmaceuticals in its present form.

Response: 

Thank you very much for your generous comment. We are thrilled to hear that you agree with the findings in the review and that you suggested the review for publication in its present form. We would just like to point out that we made several changes to the article. Per the request of a reviewer, we changed the format of the paper into a more narrative format by removing Methods and Results sections. However, we did not delete any major writings because the writings. Methods and Results sections were absorbed into a new sub-heading “Selection of the Studies”. In Selection of the Studies, we explained how we collected studies which we used to write this review but in a more narrative manners. Again, we are grateful that you took the time to read our paper.

Reviewer 2 Report

Comments and Suggestions for Authors

The submitted review article summarizes data on sirtuins 1-7 in hearing protection. The organization of the manusript is quite difficult to understand: the first part (abstract, introduction , methods, results) seems more related to a Systematic review-meta-analysis with a missing statistic test; discussion is organized as a review article and provides a very detailed overview on the 7 sirtuins in hearing loss/protection, but lacks the general background that usually characterizes narrative reviews. This organization is quite unusual for a narrative review article. Thus I suggest to adjust the article,   starting from a clear definition of the article type. 

Comments on the Quality of English Language

 Minor editing of English language required

Author Response

Comment 1:

The organization of the manuscript is quite difficult to understand: the first part (abstract, introduction , methods, results) seems more related to a Systematic review-meta-analysis with a missing statistic test; discussion is organized as a review article and provides a very detailed overview on the 7 sirtuins in hearing loss/protection, but lacks the general background that usually characterizes narrative reviews. This organization is quite unusual for a narrative review article. Thus I suggest to adjust the article, starting from a clear definition of the article type. 

Response 1:

Thank you very much for your comment. We are sorry to hear that you found the paper difficult to understand. Your concern is well-founded and we understand where we went wrong. Per your recommendation, we adjusted the article in many ways. First, the paper now states that it is a literature review, not a systematic review. This statement can be seen in Abstract and in the last paragraph of Introduction. We believe that the format of our paper is a literature review which gives readers information on the topics, discusses each paper in depth, and identifies gaps of knowledge. We also adjusted the headings. Introduction-methods-results-summary is now gone, as they are appropriate for a systematic review. Now we have the structure of Introduction-Selection of studies-Discussion-Summary, which is commonly seen in narrative reviews. We were considering whether we should delete the methods and results sections all together, but we decided against it. We integrated some important aspects of the methods and results sections into the new headings. We believe that informing the readers about the literature search method is important because it shows that we were unbiased in selecting papers and our writing could be an important framework for writing future reviews. In a new sub-heading 2.3, we introduce the experimental data we are looking for in papers and explain how scientists in the fields of sirtuins and auditory systems measure the expression of proteins and sensitivities of hearing at different frequencies respectively. Together with information on the molecular function of sirtuins in Introduction, we believe that we provided the general background information necessary for hearing research peers.

We would like to further mention that we deleted sub-headings such as Visual display of results/Heterogeneity among study results/Assessments of risk of bias due to missing results/reporting biases, because they are appropriate for a systematic review with PRISMA guidelines. Instead, heterogenous results and biased studies were discussed in Discussion section. 

We also detected typos and writing format errors detected during the review process, and addressed them. 

Again, we would like to thank you for taking the time to read the paper. We are grateful that you pointed these out. 

Reviewer 3 Report

Comments and Suggestions for Authors

The manuscript is a review done at a very good level. I have no critical comments regarding the design, organization of the text, or interpretation of the data provided. The purpose and novelty are clearly indicated in the Introduction, all sections are logical and understandable. The issues discussed are visualized in tables and figures. The low level of the text's identity testifies to the great and high-quality work of the authors. It is a really good work, and I recommend to accept it for publication. The only thing I would like to point out is that an editorial edit of the text is necessary.

Comments on the Quality of English Language

In my opinion, everything is fine. It may be necessary to edit the text in terms of dots and typos.

Author Response

Comment 1: The only thing I would like to point out is that an editorial edit of the text is necessary. In my opinion, everything is fine. 

Response 1: Thank you very much for your generous comments. We are delighted to hear your praises.

Comment 2:  I have no critical comments regarding the design, organization of the text, or interpretation of the data provided.

Response 2: Thank you very much for your comment. We would like to inform you that we made a structural change to the article per the request of a reviewer. We changed the format of the paper into a more narrative format by removing Methods and Results sections. However, we did not delete any major writings because the writings in Methods and Results sections were absorbed into a new sub-heading “Selection of the Studies”. In Selection of the Studies, we explained how we collected studies which we used to write this review. We also used tones which are more appropriate for a narrative review, and avoided phrases which are more appropriate for a systematic review.

Comment 3: It may be necessary to edit the text in terms of dots and typos.

Response 3: We found and corrected a number of typos and writing format errors. We are disappointed that we did not find them before submission. Again, thank you very much for taking the time to read the paper.